# The effectiveness of Non-pharmaceutical interventions in reducing the COVID-19 contagion in the UK, an observational and modelling study

Giorgos Galanis[1]☉, Corrado Di Guilmi[2‡], David L. Bennett[3‡], Georgios Baskozos[3☉]*

1 Institute of Management Studies, Goldsmiths, University of London, London, United Kingdom,
2 Department of Economics, University of Technology Sydney, Sydney, Australia, 3 Neural Injury Group, Nuffield Department of Clinical Neuroscience, John Radcliffe Hospital, University of Oxford, Oxford, United Kingdom

☉ These authors contributed equally to this work.
‡ CDG and DLB also contributed equally to this work
* georgios.baskozos@ndcn.ox.ac.uk

**Data Availability Statement:** All data is publicly available from https://covid19.who.int/?gclid=CjwKCAjwltH3BRB6EiwAhj0IUD_

## Abstract

Epidemiological models used to inform government policies aimed to reduce the contagion of COVID-19, assume that the reproduction number is reduced through Non-Pharmaceutical Interventions (NPIs) leading to physical distancing. Available data in the UK show an increase in physical distancing before the NPIs were implemented and a fall soon after implementation. We aimed to estimate the effect of people's behaviour on the epidemic curve and the effect of NPIs taking into account this behavioural component. We have estimated the effects of confirmed daily cases on physical distancing and we used this insight to design a behavioural SEIR model (BeSEIR), simulated different scenaria regarding NPIs and compared the results to the standard SEIR. Taking into account behavioural insights improves the description of the contagion dynamics of the epidemic significantly. The BeSEIR predictions regarding the number of infections without NPIs were several orders of magnitude less than the SEIR. However, the BeSEIR prediction showed that early measures would still have an important influence in the reduction of infections. The BeSEIR model shows that even with no intervention the percentage of the cumulative infections within a year will not be enough for the epidemic to resolve due to a herd immunity effect. On the other hand, a standard SEIR model significantly overestimates the effectiveness of measures. Without taking into account the behavioural component, the epidemic is predicted to be resolved much sooner than when taking it into account and the effectiveness of measures are significantly overestimated.

## Introduction

The spread of severe acute respiratory syndrome coronavirus 2 (SARS-CoV-2), the virus responsible for COVID-19 has led to more than 9,277,214 confirmed cases and more than

3,032,124 deaths as of 21/04/2021 [1]. Apart from the health-related implications, COVID-19 has been affecting almost every aspect of people's lives and these effects have been unequally distributed [2].

Governments have resorted to health policies known as Non-Pharmaceutical Interventions (NPIs), aimed to reduce the average number of contacts between individuals (physical distancing). While during the first period of the pandemic where no vaccines had been available, NPIs had been the only available measure in reducing the spread of the virus, since a number of vaccines became available in a number of countries, NPIs are being used in parallel to vaccination [3–5].

A number of studies have explored the effects of NPIs on the contagion dynamics of COVID-19. [6–11]. One of the focuses of relevant models is the estimation of the effects of NPIs on the rate of spread of the epidemic, captured by the reproduction number, which differs across countries [12] and this can be due to a number of social and economic conditions [13, 14]. Hence, the effectiveness of the different policies depends on how the various measures reduce this parameter. In order to be able to capture the level of this effect, it is necessary to estimate the value of the reproduction number, which in standard compartmental models is assumed to be initially close to constant and changes as a response to active NPIs [12].

However, data which capture mobility levels of individuals show that in a number of countries, including the UK, people took physical distancing measures before governments imposed NPIs. This behaviour, fits well with the insights of models, which take into account behavioural changes over and above NPIs [6, 15] such that the effective reproduction number becomes (at least partly) endogenous. Nonetheless, these works are theoretical and have not been applied to data sets related to COVID-19 so far.

The purpose of our study was threefold: (i) assess the influence of NPIs on physical distancing in the UK, taking explicitly into account the physical distancing behaviour due to observed cases; (ii) extend the standard SEIR model in order to incorporate individuals' physical distancing behaviour using the analysis from (i); and (iii) use this extended model to study the effectiveness of the NPIs, including the level and timing of NPIs imposed and the possible effects of lifting the relevant measures.

## Methods

### Data and statistical analysis

In order to assess the influence of both NPIs and the observed information, we analysed the correlation between individuals' mobility levels and the number of daily confirmed cases of the previous day as reported in the WHO dashboard [1] for three different periods: (a) up to the point when physical distancing advice was given (b) between this advice and enforceable lockdown, and (c) after lockdown. Enforceable lockdown includes NPIs ranging from the closure of public spaces, transportation hubs and shops to forbidding interactions with people outside one's household and ban any unnecessary travel. Following Buckee et al. (2020) [16], we created an aggregated data time series for physical distancing in the UK using data from Google's "COVID-19 Community Mobility Reports" [17]. Data in Google's Community Mobility Reports has undergone differential anonymisation and does not contain any personally identifiable information. Data are generated by aggregation people that have turned on their "location history" setting. Data points are expressed as percentage differences from baseline and validated to be in the expected range of –100 to 100.

Data shows the changes in mobility in six different categories:

1. *Retail and recreation*, reporting the mobility trends for places such as restaurants, shopping centres, libraries and cinemas.

2. *Supermarket and pharmacy*, capturing the trends for places such as supermarkets, food warehouses and pharmacies

3. *Parks*, which shows the mobility trends for places like parks, public beaches, plazas and public gardens.

4. *Public transport*, which shows the mobility trends for places that are public transport hubs, such as underground, bus and train stations.

5. *Workplaces*, capturing mobility trends to places of work

6. *Residential*, which shows mobility trends for places of residence.

We noted that not all of the above categories are relevant for measuring levels of physical distancing, which on one hand are related to both NPIs and to individuals' behaviour, while on the other are relevant for the contagion dynamics. For this reason, we used the categories "Workplaces", "Public transport", "Retail and recreation". We defined as mobility $m_t$ at time $t$, a weighted average of these three categories. In order to calculate the different weights, we first matched these categories with the relevant ones from the national travel survey, 2018 [18] which includes the following travel categories: Business, Education, Escort education, Shopping, Personal Business, Visiting friends at private home, Visiting friends elsewhere, Entertainment / public activity, Sport, Holiday, Day trip, Other.

We matched the National Travel Survey categories "Holiday", "Day trip", "Entertainment / public activity", "Shopping", "Visiting friends elsewhere (than home)" to the "Retail and Recreation" mobility trend; "Commuting", "Business" and "Personal Business" to the "Workplaces" mobility trend. Additionally, we hypothesised that the "Public Transport / Transit" mobility trend uniformly influences both the above trends. We then computed the relative weights of the above activities with regards to the total activities and mapped these weights to the three mobility categories. This gave us relative weights of 0.38 for "Retail and Recreation", 0.29 for "Workplaces" and 0.33 for "Public Transport/Transit". We observe that mobility over time resembles a logistic distribution and the same is true for new confirmed cases ($c_t$), (Fig 1).

The gold line represents the 7-day rolling mean of the new confirmed cases per day. Values are scaled to the range of 0–100. The green line represents the weighted average of the percentage drop of mobility trends from baseline. Vertical dashed lines show the timing of various Non-Pharmaceutical Interventions. Mobility data has been downloaded from google mobility trends and is publicly available, [17] new daily cases has been downloaded from the WHO dashboard [1] and is publicly available.

We observed two periods: one with a sharp decrease in mobility and one with a slight increase. The plots of the confirmed cases follow a very similar pattern but in opposite directions. The different NPIs seem to have an effect of the slope of both lines. The fall in confirmed cases happens around 14 days after the NPIs have been introduced and a big fall in mobility has taken place.

Based on this observation, it is reasonable to assume that there is a linear relationship between the two data series and that the available information in one period affects the decision for the next, which means that $m_t$ is a linear function of $c_{t-1}$ ie.

$$m_t = \lambda_0 c_{t-1} + \lambda_1,$$

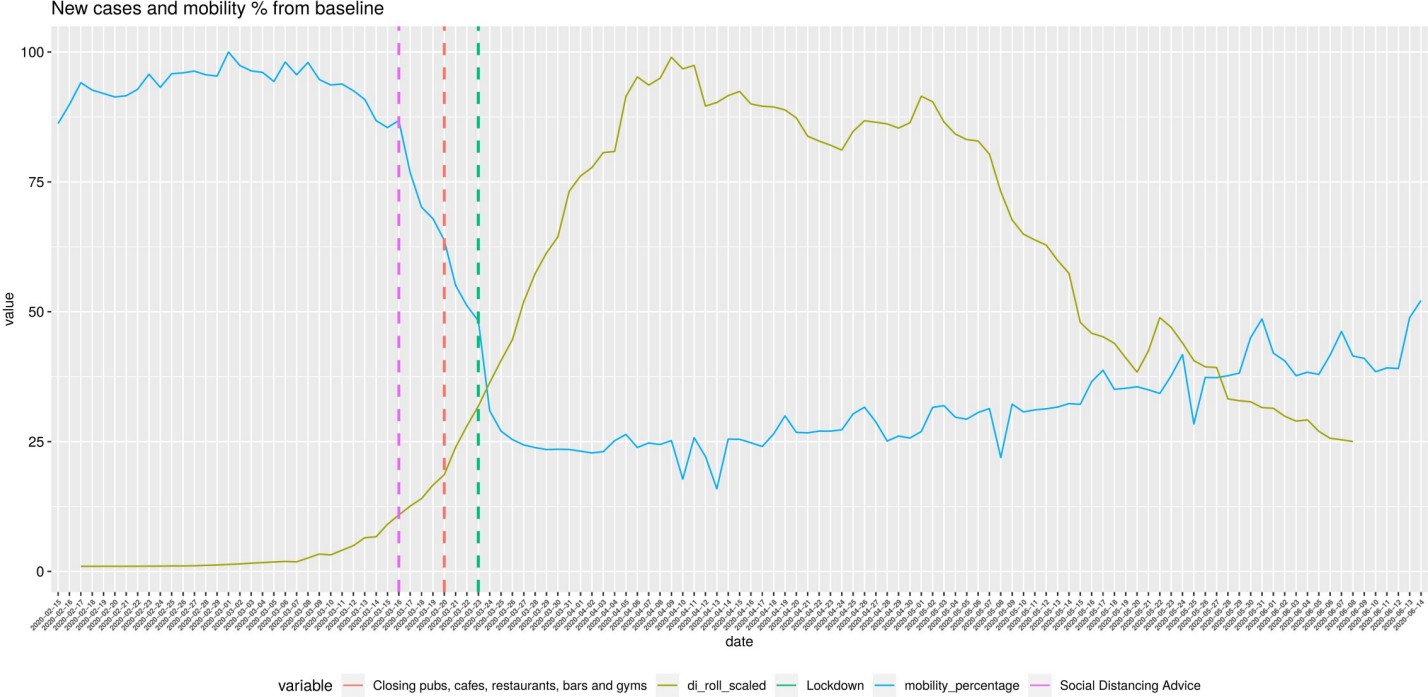

**Fig 1. Mobility and new confirmed cases for the UK 15-02-2020: 14-06-2020.**

where $\lambda_0$ captures the effect of daily confirmed cases and $\lambda_1$ other influences including the effects of NPIs.

However, given that also NPIs affect mobility, we tested this hypothesis for three different periods: before advice, between advice and lockdown, and after lockdown. This hypothesis is supported by very high and significant correlation between the two variables in all three different periods (S1 Fig).

This observation highlights the fact that NPIs are not the only factors which influence mobility which in turn is related to physical distancing levels and the reproduction number of COVID-19. Behaviour should therefore be taken into account in relevant models and policy simulations.

## Behavioural SEIR Model

The key variable informing NPIs is the reproduction number, which is the fraction of the transmission rate of the epidemic ($\beta_t$) over the recovery rate of infected individuals ($\gamma$). The daily transmission rate (and hence the basic or effective reproduction number) directly depends on the number contacts per individual, which means that due to the assumption that mobility is a proxy of daily number of contacts we can express $\beta_t$ as

$$\beta_t = Pm_t = \zeta c_{t-1} + \varepsilon \tag{1}$$

where $P$ is the probability of getting infected if susceptible, $m_t$ is the mobility as defined above $\varepsilon = P\lambda_0$ and $\zeta = P\lambda_1$ capture the relative importance of the behavioural component related to observed infections and the NPIs respectively, such that for $\zeta = 0$ only NPIs would affect the level of $\beta_t$. Note that $\zeta$ and $\varepsilon$ are the slope and intercept of the linear model fitting the data as described in the previous section.

We divided the population of individuals according to the infection status into susceptible ($S_t$), exposed ($E_t$), infected ($I_t$) and removed ones ($R_t$) such that

$$S_t + E_t + I_t + R_t = N. \tag{2}$$

Susceptible subjects might get infected when they contact an infectious individual and if infected, they enter the exposed compartment before the infected one. Following Prem et al. (2020) [8], the infected individuals are split into two further groups where the first group is symptomatic and clinical ($I_t^C$) and the second asymptomatic and subclinical ($I_t^{SC}$). The first group is a fraction $\rho$ of the total infected and the second is 1 - $\rho$ of the total.

Accordingly, the evolution of the infection is given by the following set of equations

$$S_{t+1} = S_t - \beta_t S_t I_t / N, \tag{3}$$

$$E_{t+1} = (1 - \alpha)E_t + \beta_t S_t I_t / N, \tag{4}$$

$$I_{t+1}^C = \rho\alpha E_t + (1 - \gamma)I_t^C, \tag{5}$$

$$I_{t+1}^{SC} = (1 - \rho)\alpha E_t + (1 - \gamma)I_t^{SC} \tag{6}$$

$$R_{t+1} = R_t + \gamma I_t, \tag{7}$$

with $I_t = I_t^C + I_t^{SC}$, where $\beta_t$ captures the daily transmission rate of COVID-19 (as above), $\alpha$ is the parameter related to the incubation with $1/\alpha$ being the average incubation period (in days) and $\gamma$ is the daily probability that an infected individual becomes removed (as above). Note that the confirmed daily cases $c_t$ refer to only the symptomatic, which means that $c_t = \rho\alpha E_{t-1}$. hence Eq (1) can be expressed as

$$\beta_t = \zeta\rho\alpha E_{t-2} + \varepsilon \tag{8}$$

We note that intuitively $\zeta$ should be negative such that an increase in confirmed cases leads to an increase in physical distancing practices, hence to a reduction in $\beta_t$.

Eqs (2)–(8) constitute the baseline Behavioural SEIR (BeSEIR) model based on UK data. The last equation captures the behavioural part of the compartmental model based in the UK and may differ for other countries or other epidemics as the linear relationship which we observed may not hold in other jurisdictions or countries.

## Results and discussion

### Results

We simulated the contagion dynamics in the UK and the different effects of policies over two periods of 200 and 300 days, respectively, with the demographic changes being ignored, hence keeping the total number of individuals as constant. For Eqs (3)–(7), we used values from relevant published works. We used the data from a counterfactual SEIR model with $\beta = \beta_0$, $\gamma = 0.133$, $\rho = 0.4$ (Prem et al. 2020) [8], N = 50000000 (roughly the number of adults in the UK [19], and started with 1000 people exposed at t = 0. We started our simulations with a seed of 1000 people exposed on Day 0. Genetic analysis has showed 1356 transmission lineages of COVID-19 in the UK [20].

We calibrated the constants of Eq (8) as follows:

For $E_{t-2} = 0$, we get $\beta_0 = \varepsilon$. Eq (1) can be expressed as $R_0\gamma = Pm_0$, which means that for $m_0 = 1$ (equivalent to 100% mobility levels) and given values of $\gamma$ and $R_0$ we can find $P$ and $\beta_A$, the value of the reproduction number at the time when the NPIs are introduced.

From Flaxman et al. (2020) [12], we set $R_0 = 3.8$, which gives $\beta_0 = \varepsilon = R_0\gamma = 0.5$. We observed that the UK government started implementing measures when new cases per day were 409 [1]. Using the counterfactual BeSEIR model, we chose the time step with the closest value, which was 441 on day 24. Hence, for $\rho\alpha E_{t-2} = 441$, $\beta_A = 441\zeta + \beta_0$

which gives $\zeta = (\beta_A - \beta_0)/409 = -0.00016$.

We then considered the peak in new clinical cases per day as the time of the maximum physical distancing and $m_t = 0.2$ and thus the lowest value of the effective reproduction number. Based on this we calculated the value of $\beta_t$ on that day (at t = 104), which we call $\beta_B$ and is 0.10. Using this value, we calculated the parameters of Eq (8) after the measures. Call these parameters $\zeta' = -0.0001734341$ and $\varepsilon' = 0.1769653$. Using these values, our extended BeSEIR model was able to reproduce the dynamics of contagion in the UK (Fig 2A and 2B).

We compared the simulation results of our model with a standard SEIR model and found that the latter model predicts a number of infections both cumulative (Fig 2C) and per day, much higher than the BeSEIR (Fig 2D and 2E). The difference regarding infections is several orders of magnitude, which highlights the importance of taking into account behavioural factors.

We compared the effectiveness of NPIs between our model and the standard SEIR. As expected in the standard SEIR, the cumulative number of infected individuals is lower than in the BeSEIR (Fig 3A). As in the previous case, the cumulative number differs by several orders of magnitude. We noticed that the reproduction number on the standard SEIR is only affected by NPIs (Fig 3B).

We tested the impact on the total number of infected individuals and the maximum confirmed daily cases of the delay of (i) implementing the measures (ii) and lifting restrictions. This allowed us to compute the cost in terms of lockdown days in order to reach the same reproduction number. The next graph shows the dynamics of key variables compared to a hypothetical situation when the same measures had been taken 7 days earlier than the actual date of intervention.

We noticed that, while the timing of measures has an important impact on the number of infected individuals (both daily and cumulatively, Fig 4A–4C), the reproduction number is reduced relatively less compared to the scenario where the measures are taken later (Fig 4D). This highlights that all other factors being equal, it may be optimal to have a relatively higher reproduction number with a lower number of infected rather than the opposite. This is due to the fact that the number of infected individuals at any point in time depends both in the reproduction number and the number of infected in the previous period. Hence a later intervention would require a higher reduction in the reproduction number to have the same reduction in infections to an earlier one.

Interestingly, we noticed that the maximum number of infected individuals at any point in time using the BeSEIR model is significantly and much more realistically lower than the standard SEIR (Fig 4C and 4D), where it also takes longer for the number of infected to be reduced.

We tested the impact of lifting the measures earlier rather than later and also compared this to the hypothetical case of earlier timing of NPIs. As expected, the most efficient policy would be to both delay lifting physical distancing measures and implementing NPIs early (Fig 5). We noticed that taking the timing regarding on when the measures are lifted plays a less important role (assuming that there will be a lift) compared to the timing of imposing the measures.

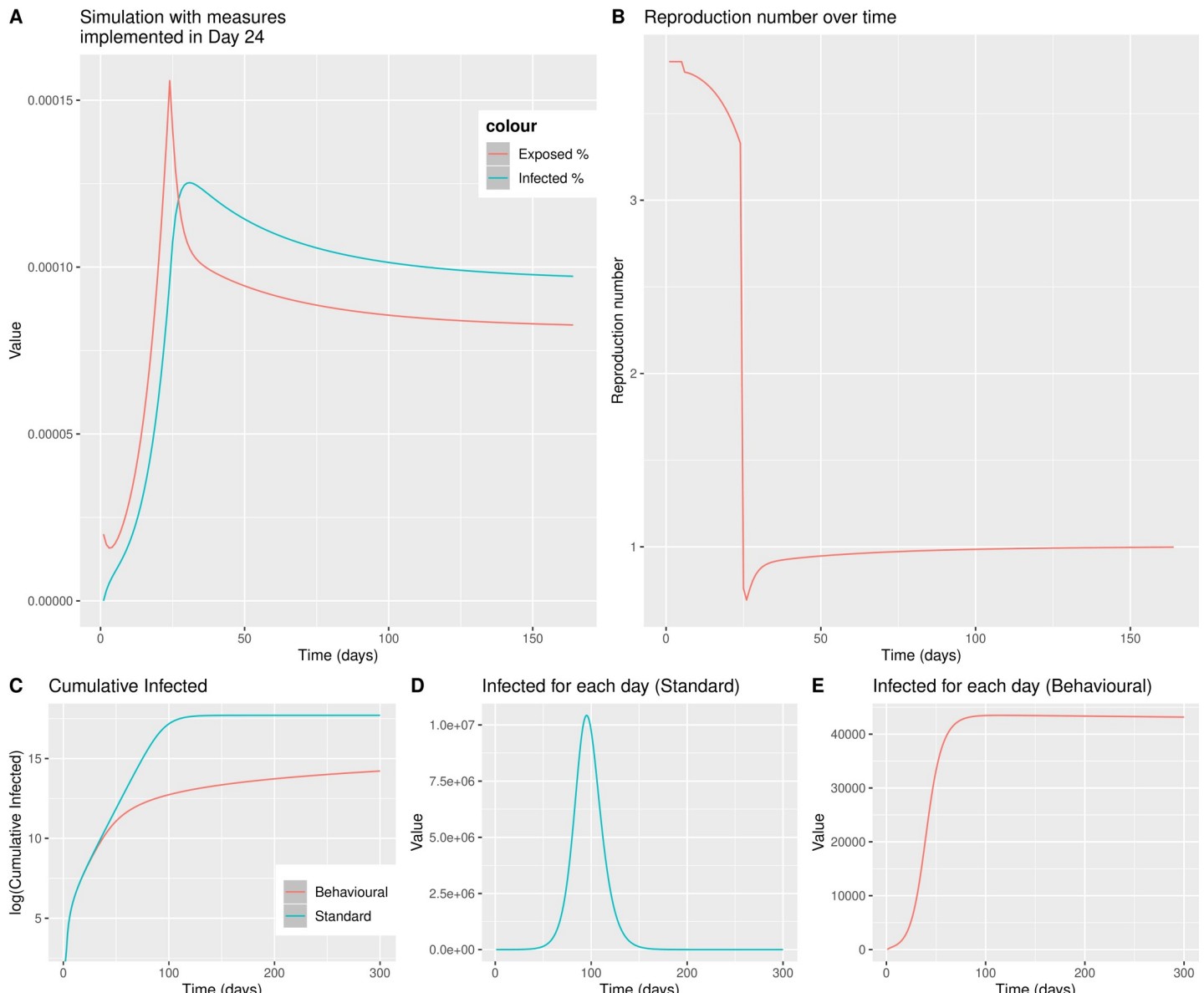

**Fig 2. Behavioural SEIR model simulation for the UK adult population, with measures implemented on Day 24, when the number of new observable cases per day is similar to what was in the UK on 16-03-2020.** Panel A shows the percentage of Exposed and Infected (both clinical and sub-clinical) individuals per day. Panel B shows the reproduction number of the disease (Rt) over time. Panel C shows the cumulative infected people as predicted with the standard SEIR model–blue line vs the BeSEIR model–red line. Panel D shows the infected people at any time point as predicted by the standard SEIR. Panel E shows the infected people at any time point as predicted by the behavioural SEIR. $R_0 = 3.8$, $\alpha = 0.15625$, $\gamma = 0.1331221$, starting seed = 1000 individuals exposed on Day 0.

## Discussion

In this paper argued that in order to be able to evaluate the effectiveness of NPIs, it is necessary to explicitly take account of the behavioural change with regards to physical distancing due to both the relevant NPIs and independent individual choices. If the behaviour of individuals is not taken into account, the levels of physical distancing without measures can be underestimated and similarly the effectiveness of measures can be overestimated. Hence, incorporating an autonomous element of physical distancing can also increase the accuracy of modelling predictions.

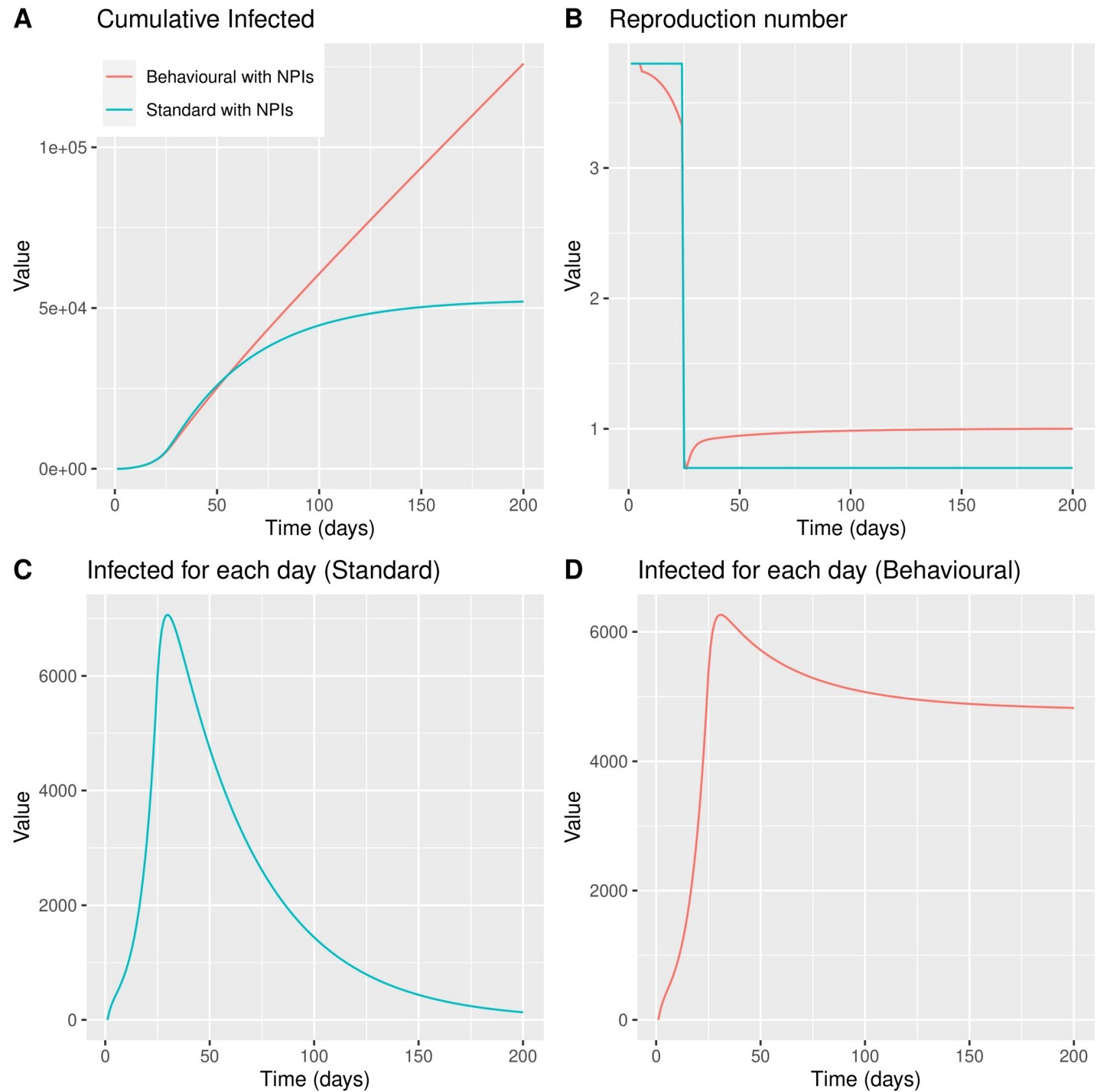

**Fig 3. BeSEIR vs standard SEIR with NPIs implemented on Day 24.** Panel A shows the cumulative infected people as predicted with the standard SEIR model–blue line vs the behavioural SEIR model–red line. Panel B shows the reproduction number over time. Panel C shows the infected people at any time point as predicted by the standard SEIR. Panel D shows the infected people at any time point as predicted by the behavioural SEIR.

Using aggregate mobility data for the UK, we observed that individual mobility levels had been reducing before the measures were taken and have been increasing even before the announcement of relaxation of the measures. We tested whether information regarding

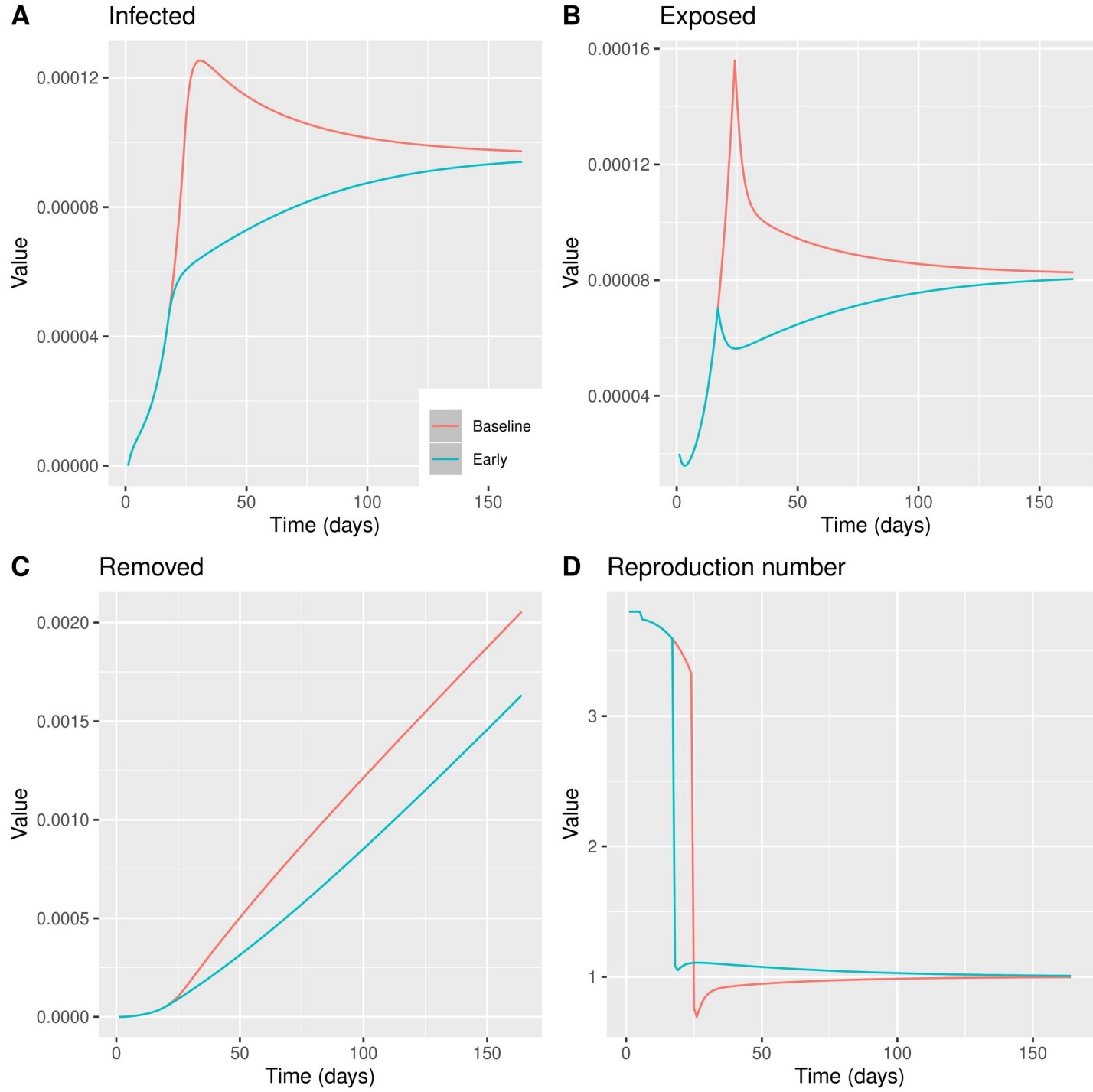

**Fig 4. BeSEIR model comparing the effect of measures taken on Day 24, baseline model–red line or measures taken on Day 17, early model–blue line.** A: Percentage of infected individuals. B: Percentage of Removed individuals. C: Percentage of Exposed individuals. D: Reproduction number $R_e = \beta_t / \gamma$.

confirmed cases can explain the changes in mobility within the different periods of NPIs. In order to also take into consideration, the effects of policies, we considered three distinct periods: before advice, between advice and lockdown, and after lockdown. We found high correlation in all three periods, which confirms the fact that people have been making physical

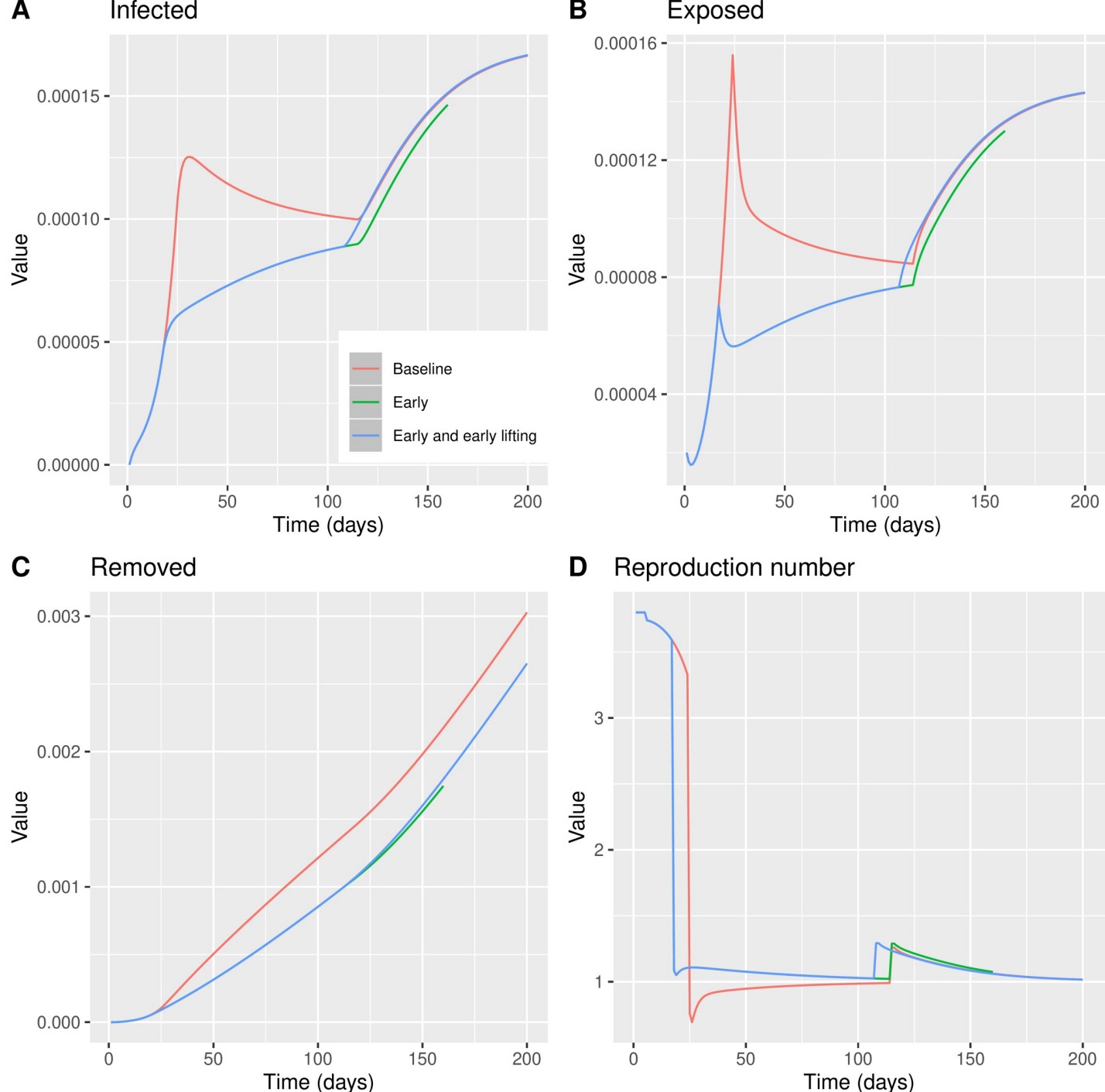

**Fig 5. BeSEIR model comparing the effect of measures taken on Day 24 and lifted on Day 114, baseline model–red line, measures taken on Day 17 and lifted on Day 114, early model–green line, or measures taken on Day 17 and lifted on Day 107, early model and early lifting–blue line.** A: Percentage of infected individuals. B: Percentage of Removed individuals. C: Percentage of Exposed individuals. D: Reproduction number $R_e = \beta_t/\gamma$.

distancing choices using the available information regarding the number of cases which are also assumed to be correlated to the number of deaths. We note that the number of cases reported, at the early phases of the epidemic at least, was a gross underestimate of the real

cases. However, the number of deaths has been used in order to infer the number of cases by imposing a somewhat arbitrary death rate. Given this as long as the number of deaths is a fraction of the number of cases the outcome of our BeSEIR model will not change and in addition we acknowledge that people make decisions based on imperfect information.

Our results not only confirm that individual behaviour should be taken into account but also provides a functional form that can be used in models which have similar assumptions regarding physical distancing behaviour [6, 7, 15]. Other studies modelling the COVID-19 pandemic taking into account government interventions implicitly assume a behavioral change as a function of NPIs [21–23], whereas our study links the behavioral component to the available information and the current state of the progression of the pandemic. Furthermore, this type of empirical exercise can be replicated across countries to analyse the relative role of social and economic conditions in physical distancing practices along the lines discussed in [13, 14].

This observation raised two policy related questions with regards to the timing of making the interventions and the time of lifting these. Given that individuals react autonomously, policies are less effective compared to a situation where individuals do not act independently. However, it is not clear how much this behaviour would impact the overall results.

We showed that when the level of daily infections is relatively low, more strict measures are required in order to achieve high levels of physical distancing. This means that the same level of measures may be less effective in reducing the reproduction number if these are imposed earlier rather than later. But importantly, this does not mean that early measures are less effective in reducing the overall number of infections. NPIs which are imposed even a week earlier can have an important impact in the reduction of infections.

This finding may give an explanation about the initial spread of the epidemic in countries that were hit first. Given the high uncertainty and the much lower volumes of information the endogenous behavioural component we describe couldn't have a significant effect in these countries. On the other hand, quickly introduced NPIs in the form of enforceable lockdowns were the only way to reduce the spread of the epidemic.

## Conclusion

In order to be able to assess the effects of the different policies, we first extended the standard SEIR model to a BeSEIR one and calibrated it using epidemiological data from previous relevant studies. Our results highlight two issues. First, not taking into account the fact that individuals also react themselves over and above NPIs may lead to very misleading projections with regards to the effectiveness of measures. Second, the fact that even though the reproduction number is a relevant variable for policy purposes, it is not necessarily a measure of success of NPIs. A higher reproduction number with less active cases can be preferable to the opposite can be less challenging to the capacity of a given health system. Of course, the basic reproduction number of the disease as defined by the biological features of the SARS-COV-2 virus is still important as it is predictive of the epidemic curve in the absence of NPIs or changes in human behaviour.

$$R_0 R_e = \beta_t/\gamma R_e = \beta_t/\gamma$$

## Supporting information

**S1 Fig. Scatterplots showing the relationship between the mobility percentage drop from baseline and the logit transformation of new COVID-19 cases per day in the UK.** A regression line is drawn in black with the corresponding 95% confidence interval in grey. We also show the Pearson's correlation coefficient—"R" and the corresponding p.value—"p" assessing

the significance of the correlation coefficient estimate. A: The relationship before the advice for maintaining physical distancing. B: The relationship after the advice for physical distancing but before enforceable lockdown. C: The relationship after lockdown.
(TIF)

## Author Contributions

**Data curation:** Georgios Baskozos.

**Writing – original draft:** Giorgos Galanis, Georgios Baskozos.

**Writing – review & editing:** Corrado Di Guilmi, David L. Bennett.

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
