## [Decision Letter · Decision Letter 0]

16 Sep 2021

PONE-D-21-13349The effectiveness of Non-Pharmaceutical Interventions in reducing the COVID-19 contagion in the UK, an observational and modelling study.PLOS ONE

Dear Dr. Baskozos,

Thank you for submitting your manuscript to PLOS ONE. After careful consideration, we feel that it has merit but does not fully meet PLOS ONE’s publication criteria as it currently stands. Therefore, we invite you to submit a revised version of the manuscript that addresses the points raised during the review process.

ACADEMIC EDITOR: Please modify the manuscript==============================

We look forward to receiving your revised manuscript.

Kind regards,

Prasenjit Mitra, MD, CBiol, MRSB, MIScT, FLS, FACSc, FAACC

Academic Editor

PLOS ONE

Journal Requirements:

3. Please upload a copy of Supporting Information Figure 1 which you refer to in your text on page 4.

Reviewers' comments:

Reviewer's Responses to Questions

**Comments to the Author**

1. Is the manuscript technically sound, and do the data support the conclusions?

Reviewer #1: Yes

2. Has the statistical analysis been performed appropriately and rigorously? 

Reviewer #1: Yes

3. Have the authors made all data underlying the findings in their manuscript fully available?

Reviewer #1: Yes

4. Is the manuscript presented in an intelligible fashion and written in standard English?

Reviewer #1: Yes

5. Review Comments to the Author

Reviewer #1: The authors would need to revise the manuscript in line with the following;

Introduction:

The statement on the second paragraph may need to be revised to reflect the present day realities of availability of vaccines but with problems of vaccine hesitance.

Methods:

The authors may need to state how the data from Google’s “COVID-19 Community Mobility Reports were validated and abstracted.

Discussion:

The discussion section will need to revised from just discussing the findings of this study solely to brining into context findings of others studies

Conclusion:

The authors may consider summarising the conclusion as it is too lengthy in this present form

6. PLOS authors have the option to publish the peer review history of their article (what does this mean?). If published, this will include your full peer review and any attached files.

Reviewer #1: **Yes: **Tolulope Olumide Afolaranmi

---

## [Author Response · Author response to Decision Letter 0]

27 Sep 2021

Dear editors,

Thank you for giving us the opportunity to submit a revised manuscript to PLOS One. Below is a detailed response to all the points raised by the reviewer. Our point by point response is in bold. In addition to the points discussed below we have edited the manuscript to be fully concordant with the formatting templates provided, amended the funding statement to reflect all funding sources, uploaded the missing “Supporting Information Figure 1” and reviewed our reference list. We thank the editor for providing constructive feedback on this issues. We have taken all their comments into account and we feel that they made the manuscript stronger.

Yours sincerely, 

 Georgios Baskozos

Reviewer #1: The authors would need to revise the manuscript in line with the following;

Introduction:

The statement on the second paragraph may need to be revised to reflect the present day realities of availability of vaccines but with problems of vaccine hesitance.

Thank you very much for this comment. The reviewer is right. We have changed this statement accordingly both in the introduction and the discussion.

Methods:

The authors may need to state how the data from Google’s “COVID-19 Community Mobility Reports were validated and abstracted.

We have added a short paragraph when we present Google’s “COVID-19 Community Mobility Reports” where we discuss the expected range of values, anonymisation and data aggregation. We believe this additional information shows how data were validate to be within the expected range and provided full details on data collection and aggregation.

Discussion:

The discussion section will need to revised from just discussing the findings of this study solely to brining into context findings of others studies

We have added a short paragraph which relates our work and results with other closely related studies and added the appropriate references. We have also moved part of our conclusion to the discussion section.

Conclusion:

The authors may consider summarising the conclusion as it is too lengthy in this present form

We have significantly shortened the conclusion.

---

## [Decision Letter · Decision Letter 1]

9 Nov 2021

The effectiveness of Non-Pharmaceutical Interventions in reducing the COVID-19 contagion in the UK, an observational and modelling study.

PONE-D-21-13349R1

Dear Dr. Baskozos,

We’re pleased to inform you that your manuscript has been judged scientifically suitable for publication and will be formally accepted for publication once it meets all outstanding technical requirements.

Kind regards,

Prasenjit Mitra, MD, CBiol, MRSB, MIScT, FLS, FACSc, FAACC

Academic Editor

PLOS ONE

Additional Editor Comments (optional):

Reviewers' comments:

Reviewer's Responses to Questions

**Comments to the Author**

1. If the authors have adequately addressed your comments raised in a previous round of review and you feel that this manuscript is now acceptable for publication, you may indicate that here to bypass the “Comments to the Author” section, enter your conflict of interest statement in the “Confidential to Editor” section, and submit your "Accept" recommendation.

Reviewer #1: All comments have been addressed

2. Is the manuscript technically sound, and do the data support the conclusions?

Reviewer #1: Yes

3. Has the statistical analysis been performed appropriately and rigorously? 

Reviewer #1: Yes

4. Have the authors made all data underlying the findings in their manuscript fully available?

Reviewer #1: Yes

5. Is the manuscript presented in an intelligible fashion and written in standard English?

Reviewer #1: Yes

6. Review Comments to the Author

Reviewer #1: The authors have addressed all the the review comments and the manuscript is now publication worthy.

7. PLOS authors have the option to publish the peer review history of their article (what does this mean?). If published, this will include your full peer review and any attached files.

Reviewer #1: **Yes: **Tolulope Olumide Afolaranmi

---

## [Editor Report · Acceptance letter]

11 Nov 2021

PONE-D-21-13349R1 

The effectiveness of Non-Pharmaceutical Interventions in reducing the COVID-19 contagion in the UK, an observational and modelling study. 

Dear Dr. Baskozos:

I'm pleased to inform you that your manuscript has been deemed suitable for publication in PLOS ONE. Congratulations! Your manuscript is now with our production department. 

Kind regards, 

on behalf of

Dr. Prasenjit Mitra 

Academic Editor

PLOS ONE